# What Did You Think Would Happen? Explaining Agent Behaviour through Intended Outcomes

**Herman Yau**
CVSSP, University of Surrey

**Chris Russell** *
Amazon Web Services

**Simon Hadfield**
CVSSP, University of Surrey

## Abstract

We present a novel form of explanation for Reinforcement Learning, based around the notion of intended outcome. These explanations describe the outcome an agent is trying to achieve by its actions. We provide a simple proof that general methods for post-hoc explanations of this nature are impossible in traditional reinforcement learning. Rather, the information needed for the explanations must be collected in conjunction with training the agent. We derive approaches designed to extract local explanations based on intention for several variants of Q-function approximation and prove consistency between the explanations and the Q-values learned. We demonstrate our method on multiple reinforcement learning problems, and provide code[1] to help researchers introspecting their RL environments and algorithms.

## 1 Introduction

Explaining the behaviour of machine learning algorithms or AI remains a key challenge in machine learning. With the guidelines of the European Union's General Data Protection Regulation (GDPR) [1] calling for explainable AI, it has come to the machine learning community's attention that better understanding of black-box models is needed. Despite substantial research in explaining the behaviour of supervised machine-learning, it is unclear what should constitute an explanation for Reinforcement Learning (RL). Current works in explainable reinforcement learning use similar techniques as those used to explain a supervised classifier [22], as such their explanations highlight what in the current environment drives an agent to take an action, but not what the agent expects the action to achieve. Frequently, the consequences of the agent's actions are not immediate and a chain of many decisions all contribute to a single desired outcome.

This paper addresses this problem by asking what chain of events the agent intended to happen as a result of a particular action choice. The importance of such explanations based around intended outcome in day-to-day life is well-known in psychology with Malle [20] estimating that around 70% of these day-to-day explanations are intent-based. While the notion of intent makes little sense in the context of supervised classification, it is directly applicable to agent-based reasoning, and it is perhaps surprising that we are the first work in explainable RL to directly address this. Recent work [18] has called for these introspective abilities which they refer to as "Explainable Agency".

We present a simple addition to standard value-based RL frameworks which allows us to obtain a projection of predicted future trajectories from a current observation and proposed action. Unlike existing approaches such as [33] that predict an agent's future states by rerunning repeated forward simulations from the same initial state; we instead recover sum of the past events, weighted by the importance that the agent put on them when learning a Q-values, that lead to the agents current behaviour, and mathematically guarantee that this sum is consistent with the agent's Q-values. This allows for local interpretation of the agent's intention based on its behavioural policy. We

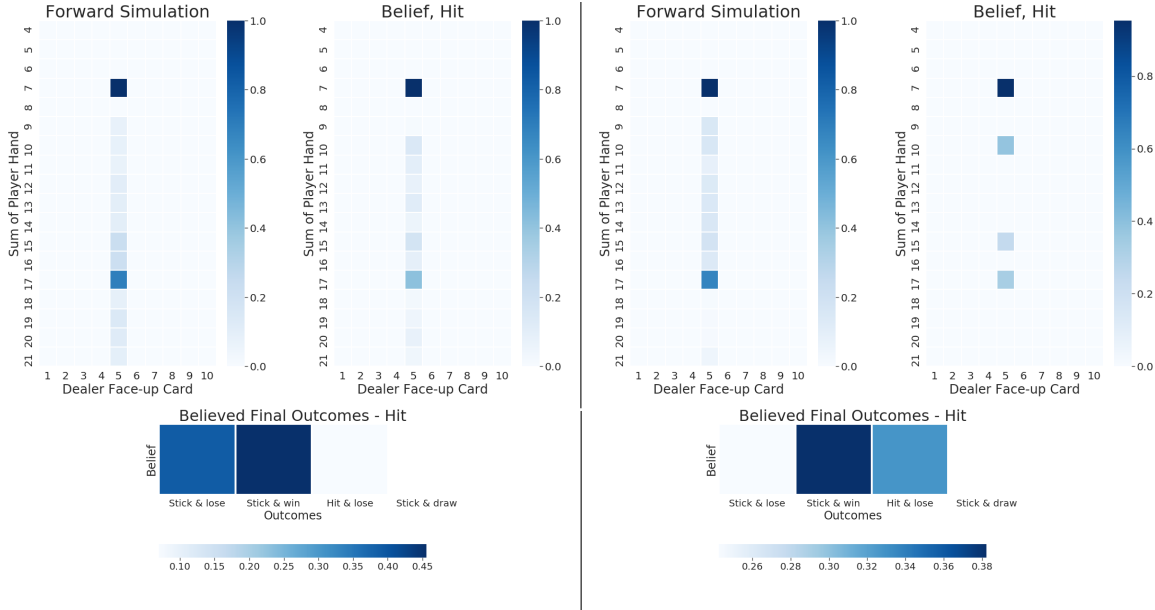

Figure 1: Understanding an agent's expected behaviour. These images show predictions of an agent's behaviour in blackjack. All plots show an initial state where the dealer shows 5, and the player's cards sum to 7 (no high aces), and the colour of each cell shows the expected number of times a state will be visited, if the agent hits. The two leftmost plots show the predicted behaviour, based on forward simulation [33], and 'belief' of a well-trained agent, while the two rightmost show the predicted behaviour and 'belief' of an underperforming agent trained on a small and fixed replay buffer. The bottom two images show the final outcomes the well-trained and underperforming agents believe will occur if they hit. This results in a conservative agent that sticks early owing to an increased belief that they will go bust from a hit.

mathematically guarantee that this is consistent with the agent's Q-values (see figure 1). We hope our method will help RL practitioners trying to understand agent behaviour for model debugging and diagnostics, and potentially lead to more reliable and trustworthy RL systems.

Mismatches between learnt Q-values and the forward simulations of [33] exist for a variety of reasons: stochasticity of the environment; mismatches between training and test environments[2]; systematic biases in the learning method [13] and the use of replay buffers (particularly hindsight experience replay (HER) [4]). Our representations highlight what is learnt by the agent at train time, rather than the behaviour of the agent at test time, allowing them to provide insights into some of these failures.

## 2 Related Works

Many approaches address interpretability for supervised machine learning. Traditional model-based machine-learning algorithms, of restricted capacity, such as decision trees [23], GLM/GAMs [14] and RuleFit [12] are considered interpretable or intrinsically explainable due to their simple nature [25]. However, the dominance of model-free algorithms and deep learning means that most approaches are not considered intrinsically explainable due to their high complexity. Unavoidably, a trade-off between interpretability and performance exists, and instead focus has switched to local, post-hoc methods of explanation that give insight into complex classifiers. Two of the largest families of explanations are: *(i)* perturbation-based attribution methods [24, 19] that systematically alter input features, and examine the changes in the classifier output. This makes it possible to build a local surrogate model that provides an local importance weight for each input feature. *(ii)* gradient-based attribution methods [28, 29, 6, 27, 26]. Here the attribution of input features is computed in a single forward and backward pass, with the attribution comprising a derivative of output against input. Fundamentally both approaches use measures of feature importance to build a low-complexity model

that locally approximates the underlying model. More recently self-explainable neural networks (SENNs) [3, 31, 2] are end-to-end models that produce explanations of their own predictions.

Despite recent progress in reinforcement learning, few works have focused on its explainability. Annasamy and Sycara [5] proposed Interpretable Deep Q Network (i-DQN), an architecturally similar model to the vanilla DQN model [21] that introduced a key-value store to concurrently learn latent state representation, and Q-value mapping for intuitive visualisations by means of key clustering, saliency and attention maps. Mott et al. [22] takes this idea further, representing the key-value store as a recurrent attention model to produce a normalised soft-attention map which tracks the steps an agent takes when solving a task. Both methods focused on identifying components of the world that drives the agent to take a particular action, rather than identifying the agents expected outcomes. As these works exploit visual attention they are only directly applicable to image-based RL problems.

Some approaches extract global policy summaries i.e. compressed approximations of the underlying policy function. Verma et al. [34] proposed Programatically Interpretable Reinforcement Learning to syntactically express policy in a high-level programming language. Topin and Veloso [32] modeled policy as a Markov chain over abstract states giving high-level explanations. Our approach gives a local description that says given a current state/action choice, what are the expected future states.

Juozapaitis et al. [15] proposed a decomposition of the scalar reward function into a vector-valued reward function for each sub-reward type. When used together for qualitative analysis, these provide a compact explanation of why a particular action is preferred over another, in terms of which sub-elements of the reward are expected to change. Our work differs in that we propose a decomposition of the Q-function over state and action space, rather than over reward sub-elements. This makes it possible for us to reason about intended future events in our explanations. Our approach is complementary to [15]; it is possible to decompose the Q-value both across state space and sub-rewards, giving detailed reasons of sub-rewards over future states.

Another work closely related to our approach is successor representation (SR) learning [9]. The SR bears some similarities to our Q-value decomposition. However, this is used specifically to support learning in non-stationary environments with distinct properties. In contrast we are interested in generating successor representations which are guaranteed to be consistent with value functions, in particular Q-learning, Monte Carlo Control, and double Q-learning.

## 3 Background

We set out the minimal background of reinforcement learning necessary to define our approach to explainable Reinforcement Learning. We formalise a reinforcement learning environment as a Markov decision process (MDP) [30]. This MDP consists of a 5-tuple $\langle \mathcal{S}, \mathcal{A}, p, r, \gamma \rangle$, where $\mathcal{S}$ is the set of all possible environment states, $\mathcal{A}$ is the set of actions, $p(s_{t+1}|s_t, a_t)$ is the transitional dynamics of the MDP, $\gamma \in [0, 1]$ is the discount factor and $r(s_t, a_t, s_{t+1}) \in \mathbb{R}$ is the reward for executing action $a_t$ in state $s_t$ arriving at state $s_{t+1}$. Reinforcement learning seeks to find the optimal policy $\pi^* : \mathcal{S} \to \mathcal{A}$ that maximise the expectation of cumulative discounted rewards over the MDP.

$$\pi^*(s) = \arg\max_{\pi} \mathbb{E}\left[ \sum_{t=0}^{\infty} \gamma^t r(s_t, a_t, s_{t+1}) \,\middle|\, s_t = s, a_t = a, \pi \right]. \tag{1}$$

Many RL approaches estimate the state-action value function $Q^{\pi}(s_t, a_t)$.[3]

$$Q^{\pi}(s_t, a_t) = \mathbb{E}_{\pi}\left[ \sum_{t=0}^{\infty} \gamma^t r_{t+1} \,\middle|\, s_t = s, a_t = a \right]. \tag{2}$$

Where we write $r_{t+1}$ as shorthand for $r(s_t, a_t, s_{t+1})$. Both value functions satisfy the Bellman equation, and are frequently estimated by Q-learning or Monte Carlo (MC) like methods.

$$Q(s_t, a_t) \leftarrow Q(s_t, a_t) + \alpha \left( r + \gamma \max_{a \in \mathcal{A}} Q(s_{t+1}, a) - Q(s_t, a_t) \right) \qquad \text{Q-learning} \tag{3}$$

$$Q(s_t, a_t) \leftarrow Q(s_t, a_t) + \alpha \left( \sum_{t'=t}^{T} \gamma^{t'-t} r_{t'} - Q(s_t, a_t) \right) \qquad \text{Monte Carlo} \tag{4}$$

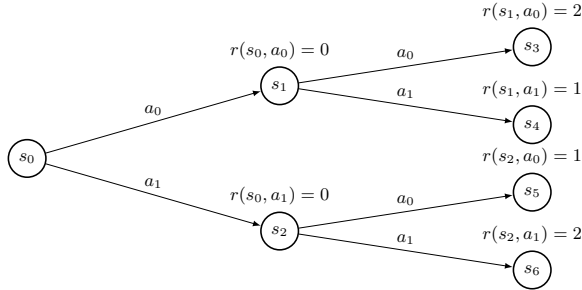

Figure 2: A simple MDP to illustrate value ambiguity in value iteration methods (see Section 4.1).

In Q-value-based deep reinforcement learning [21], the Q-function is parametrised by network parameters $\theta$. A policy is implicitly learned by computing the gradient of a loss function with experiences $e_t = (s_t, a_t, s_{t+1}, r_t)$ drawn from an experience replay $D_t = \{e_1, ..., e_t\}$. Using $\theta$ and $\theta^-$ for the behavioural network and target network parameters and assuming an $L_2$ loss, the loss function can then be written as

$$L(\theta) = \mathbb{E}_{e \sim D_t} \left[ \left( r + \gamma \max_a Q(s_{t+1}, a; \theta) - Q(s_t, a_t; \theta^-) \right)^2 \right],$$ (5)

The update equation is then simply:

$$\theta \leftarrow \theta + \alpha \nabla_\theta L(\theta).$$ (6)

## 4 Value Function Decomposition

To understand why an RL agent prefers one action over another, we want to infer the agent's implicit estimate of $p(s_{t+n}|s_t, a_t, \pi)$ for all $n > 0$ which reveals the future states used in estimating Q-values.

We do not try to understand the internal computations that led our agent to map a particular Q-value against a particular action. Instead we wish to know the expected future states that leads an agent to decide that one action in the current state would be preferable to another action.

We prove that post-hoc explanations of the form we are interested in can not be recovered from an existing agent trained using a traditional Q-learning approach; we do this by showing that the inverse problem of recovering expected future states from a Q-function is fundamentally ill-posed. Note that Q-learning is used only as an example. The same reasoning can be applied to any value-based reinforcement learning algorithm. We then derive a new algorithm that can be run alongside with an existing learning process which lets us obtain such explanations, without altering what is learnt.

### 4.1 Value Ambiguity

The Q-value function is a convenient tool for producing a loss that directs an agent's behaviour. However, it acts as a bottleneck on the information exposed. It quantifies the "goodness" (expectation of discounted rewards) for executing a given action $a$ in state $s$, without explicitly capturing *how* or *why* such a conclusion is reached. Furthermore, not only is the required information not directly available, but we also cannot in general, infer it indirectly from the Q-value.

**Theorem 1.** *Given any MDP where multiple optimal policies $\pi^*$ exist, it is not possible to produce a post-hoc interpretation due to value ambiguity.*

The proof follows immediately by contradiction. Consider the undiscounted, deterministic chain of MDP in Figure 2, where an episode starts at $s_0$. Each state has 2 actions: $a_0$ moves to the upper node, $a_1$ moves to the lower node. All rewards are 0 except for reaching the end of the chain, where $r(s_1, a_0) = r(s_2, a_1) = 2$, $r(s_1, a_1) = r(s_2, a_0) = 1$. We assume a tabular Q-learning agent trained using the $\epsilon$-greedy algorithm (with ties broken arbitrarily).

While the value functions are updated and eventually converge to optimality, there is ambiguity while learning takes place. Clearly, $Q^*(s_0, a_0) = Q^*(s_0, a_1) = 2$ and there are two optimal trajectories $\tau_{t:T}^1 = (s_0, a_0, 0, s_1, a_0, 2)$ or $\tau_{t:T}^2 = (s_1, a_1, 0, s_2, a_1, 2)$ relating to two separate policies $\pi^*$.

Clearly, in this situation it is not possible to examine a single value estimate $V^\pi(s_0)$ and infer which trajectory the agent expects to follow. Therefore we can not guarantee that any post-hoc interpretation is correct in this scenario, or in the general case. $\quad\square$

Although this is a simple example, it immediately rules out approaches to generating explanations by training an ML method to predict to future behaviour of an agent on the basis of the behaviour of previous agents with similar Q-values. The problem illustrated by our counterexample is exacerbated if we work with an extended chain of conceptually similar MDPs, as $s_3$ and $s_6$ could both contribute to the estimation of the same Q-value. Similarly, the same situation is observed for $Q^*(s_1, a_0)$ and $Q^*(s_2, a_1)$. More specifically, it is impossible to answer the contrastive question of "What different future states are expected to occur if one action is selected over another?".

## 4.2 Explaining Classical Reinforcement Learning

We describe how to augment standard RL learning methods with a simple additional process that allows us to learn a map of discounted expected states $\mathbf{H}$ (which we refer to as the *belief map*) concurrently with learning the Q-value function.

The intuition behind our approach is straightforward. While Q-value based methods (3) estimate the expected future reward summed over all discounted expected states, our approach preserves additional information, and captures the discounted expected state at the same time. By choosing update rules for $\mathbf{H}$ to match the update equations of (7) we guarantee consistency between the Q-values and the expected future states. Subsection 4.3 proves that these expected states are consistent with the learnt Q-values. We present three variations of value function decomposition. For simplicity, we assume the methods are off-policy, but our approach is directly applicable to on-policy methods.

We define $\mathbf{H}(s_t, a_t) \in \mathbb{R}^{S \times A}$ as the expected discounted sum of future state visitations in a deterministic MDP which starts by executing action $a_t$ from state $s$. As a notational convenience we use $1_{s_t, a_t} \in \mathbb{R}^{S \times A}$ to denote a binary indicator function 1 at position $s_t, a_t$ and 0 elsewhere.

**Q-Learning** Every time the Q-value estimator is updated during learning, we update the corresponding entries of $\mathbf{H}$ to maintain consistency. This is a direct adaption of Equation 3 for application to $\mathbf{H}$, that gives the update rule:

$$\mathbf{H}(s_t, a_t) \leftarrow \mathbf{H}(s_t, a_t) + \alpha \left( 1_{s_t, a_t} + \gamma \mathbf{H}(s_{t+1}, \arg\max_{a \in \mathcal{A}} Q(s_{t+1}, a) - \mathbf{H}(s_t, a_t) \right). \quad (7)$$

We update all state-action pairs in trajectory $\tau$.

**Monte Carlo Control** Adapting the update to Monte Carlo control methods is straightforward. We modify Equation 3 by replacing $1_{s_t, a_t}$ with a vector sum of discounted state visitations

$$\mathbf{H}(s_t, a_t) \leftarrow \mathbf{H}(s_t, a_t) + \alpha \left( \sum_{t=t'}^{T} \gamma^t 1_{s_t, a_t} - \mathbf{H}(s_t, a_t) \right). \quad (8)$$

In tabular settings, the equality holds when both the belief map $\mathbf{H}$ and Q-table $Q$ are zero-initialised. Under these constraints, the recovery of value function is guaranteed. If $\mathbf{H}$ is randomly initialised, the equality does not always hold, but will eventually converge to the true state visitations count, leading to an accurate representation of the agent's behaviour.

## 4.3 The Consistency of Belief Maps

We say that a belief map is consistent with a Q-table given reward map $\mathbf{R}$, if the inner product of the belief map of every state-action pair with the reward map gives the Q-value for the same state-action pair. More formally, we assume the current reward is a deterministic function of the current state-action pair, we define $\mathbf{R} \in \mathbb{R}^{S \times A}$ as a map of rewards for every state-action pair in the MDP, and say that if

$$\text{vec}(\mathbf{H}(s, a))^\top \text{vec}(\mathbf{R}) = Q(s, a) \quad \forall a \in \mathcal{A}, s \in \mathcal{S}, \quad (9)$$

then the belief map $\mathbf{H}$ is consistent with $Q$.

**Theorem 2.** *If $\mathbf{H}$ and $Q$ are zero-initialized, then $\mathbf{H}$ and $Q$ will be consistent for all iterations of the algorithm.*

*Proof.* Proof follows by induction on $i$, the iteration of the learning algorithm.

**Base case** ($i = 0$): At initialisation $\mathbf{H}_i(s, a) = \mathbf{0}$ and $Q_i(s, a) = 0$ for all possible $s$ and $a$ and the statement is trivially true.

**Inductive step (Q-learning)**: We consider iteration $i$ where we know that the theorem holds for iteration $i - 1$. We write $m_t = \arg\max\limits_{a \in \mathcal{A}} Q_{i-1}(s_t, a_t)$

$$
\begin{aligned}
\text{vec}(\mathbf{H}_i(s_t, a_t))^\top \text{vec}(\mathbf{R}) &= \text{vec}\left((1-\alpha)\mathbf{H}_{i-1}(s_t, a_t) + \alpha\left(1_{s_t, a_t} + \gamma\mathbf{H}_{i-1}(s_{t+1}, m_{t+1})\right)\right)^\top \text{vec}(\mathbf{R}) \\
&= (1-\alpha)Q_{i-1}(s_t, a_t) + \alpha\text{vec}\left(1_{s_t, a_t} + \gamma\mathbf{H}_{i-1}(s_{t+1}, m_{t+1})\right)^\top \text{vec}(\mathbf{R}) \\
&= (1-\alpha)Q_{i-1}(s_t, a_t) + \alpha r_t + \alpha Q_{i-1}(s_{t+1}, m_{t+1}) \\
&= Q_i(s_t, a_t).
\end{aligned}
$$

**Inductive step (Monte Carlo)**: We consider iteration $i$ where we know that the theorem holds for iteration $i - 1$. We use $\mathbf{H}_i$ to indicate the state of $\mathbf{H}$ at iteration $i$, and $Q_i$ the state of $Q$ at iteration $i$

$$
\begin{aligned}
\text{vec}(\mathbf{H}_i(s_t, a_t))^\top \text{vec}(\mathbf{R}) &= \text{vec}\left((1-\alpha)\mathbf{H}_{i-1}(s_t, a_t) + \alpha\sum_{t=t'} r^t \gamma^t 1_{s_t, a_t}\right)^\top \text{vec}(\mathbf{R}) \\
&= (1-\alpha)Q_{i-1}(s_t, a_t) + \alpha\text{vec}\left(\sum_{t=t'}^{T} \gamma^t 1_{s_t, a_t}\right)^\top \text{vec}(\mathbf{R}) \\
&= (1-\alpha)Q_{i-1}(s_t, a_t) + \alpha\sum_{t=t'}^{T} \gamma^t r_t \\
&= Q_i(s_t, a_t).
\end{aligned}
$$

as required. See supplementary materials for the result for double Q-learning. $\square$

## 4.4 Application to Deep Q-Learning

Much like Deep Q-learning, function approximation can be used to estimate $\mathbf{H}$. For $\mathbf{H}$ parametrised by $\theta_h$ and given agent policy $\pi_\theta$, the loss function in Equation (5) becomes:

$$
L_h(\theta_h) = \mathbb{E}_{(s_t, a_t, s_{t+1})}\left(1_{s_t, a_t} + \gamma\mathbf{H}\left(s_t, \arg\max_{a \in \mathcal{A}} Q(s_{t+1}, a; \theta); \theta_h^-\right) - \mathbf{H}(s_t, a_t; \theta_h)\right)^2. \tag{10}
$$

Then the update rule in Equation 6 becomes:

$$
\theta_h \leftarrow \theta_h + \alpha\nabla_{\theta_h} L(\theta_h). \tag{11}
$$

Here $\mathbf{H}(s_t, a_t)$ is an unnormalised density estimation. Where the state and action spaces are discrete, this can be in the form of a vector output, and where at least one is continuous using techniques such as [10], alternatively it can be approximated by pre-quantizing the state space using standard clustering techniques.

The mathematical guarantees of consistency from the previous section do not hold for deep Q-learning, with the random initialisation of the networks meaning that it fails in the base case. However, just as an appropriately set up deep Q-learning algorithm will converge to an empiric minimiser of equation (5), an appropriately setup H-learner will also converge to minimise equation (10). We evaluate the deep variants of our approach in the experimental section, and show even without theoretical guarantees, it performs effectively, providing insight into what has been learnt by our agents.

## 5 Experiments and Discussions

We evaluate our approach on three standard environments using OpenAI Gym [8] - Blackjack, Cartpole [7] and Taxi [11]. Each environments poses unique challenges. For each task we briefly

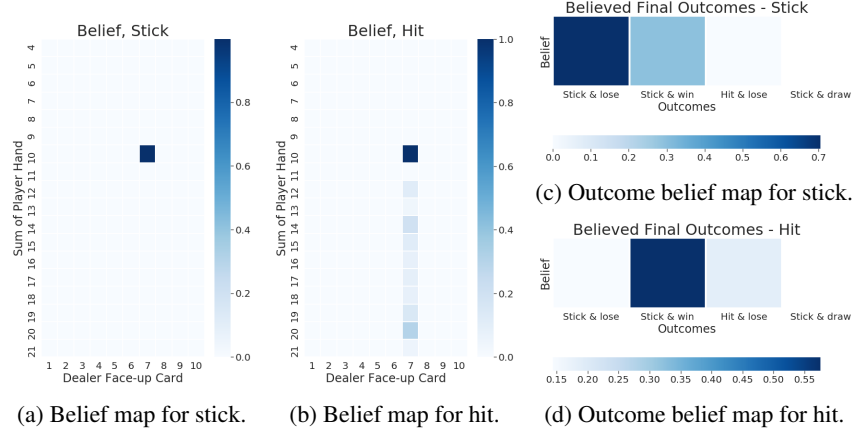

(a) Belief map for stick.     (b) Belief map for hit.     (d) Outcome belief map for hit.

Figure 3: Visualisations of belief maps for actions when player sum $= 10$, dealer card $= 7$ with no usable ace. Unreachable player states are cropped for conciseness, i.e. Player sum $< 4$ or sum $> 21$.

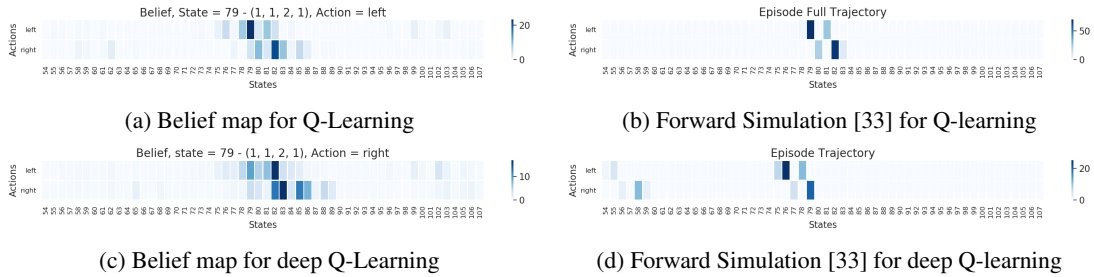

(a) Belief map for Q-Learning     (b) Forward Simulation [33] for Q-learning

(c) Belief map for deep Q-Learning     (d) Forward Simulation [33] for deep Q-learning

Figure 4: Belief maps and trajectory visualisations for cartpole simulation. Only states where the cart is in the middle are shown as it rarely drifts away to extreme positions on the left or right. A contrastive versions showing how expected state varies with action choice can be seen in the supplementary materials.

describe our assumptions and key experiment parameters. We extract an intuitive local interpretation of the RL agent's intention solely from the belief maps. We verify the correctness of our implementation in each environment using Theorem 2, and confirm equation (9) holds numerically.

We find minimal differences applying our technique to Temporal Difference and Monte Carlo approaches. For conciseness, we report here the findings from TD and the supplementary material contain MC results. RL agents always use an $\epsilon$-greedy algorithm during training.

**Blackjack** We model a simplified game of blackjack and assume (1) the shoe consists of infinite cards, (2) the only moves allowed are "hit" and "stick" and (3) we do not account for blackjacks.

The system dynamics are non-deterministic as the card drawn on each hit is random, and while the reward is stochastic due to the uncertainty in the dealer's hand. To make the reward a deterministic function of the stochastic state we create four additional states: hit and bust; stuck and won; stuck and drew; stuck and lost. The agent is trained with $\alpha = 0.1, \gamma = 1$ for $500k$ episodes. Monte-Carlo learning performed similarly to table-based Q-learning. See the supplementary results for details.

Figure 3 shows belief maps for each action and the corresponding incurred rewards when the player's hand has a sum of 10, and the dealer shows 7. There is a single point in the state space that shows a high probability in both figure 3a and 3b. This point corresponds to the current starting state, which all future trajectories must pass through. Figure 3b shows expected future events when the player hits. We observe the intention of the agent from the belief map, that it continues to hit until a high enough sum is obtained. This explains why lower values have lower density, as there are fewer paths to reach these states. The second peak at 20 is expected given that all face cards have a value of 10.

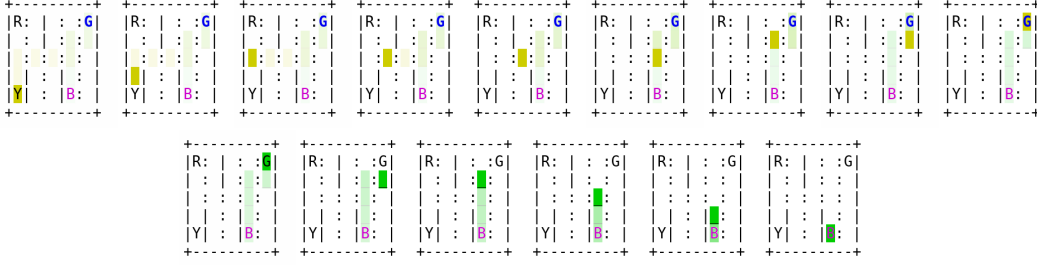

Figure 5: Varying intention of the Q-learning agent in the Taxi environment [11]. Each image represents the belief map at one point along a trajectory, with time advancing from left to right. Colour intensity denotes the confidence that the agent will visit a state. Yellow indicates the passenger is not present, whereas green means they are. 'R', 'G', 'Y', 'B', each represents a possible location of the passenger and the destination. Blue text indicates the location of the passenger if they are not in the car, and red text denotes destination.

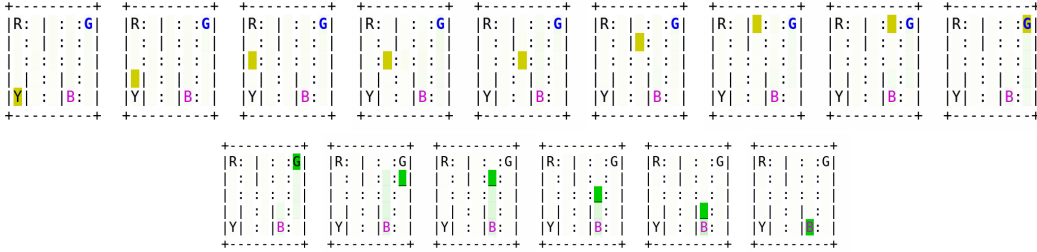

Figure 6: How the belief maps of the DQN agent vary over time in the Taxi [11].

**Cartpole** [7] is a classic continuous control problem with a 4-tuple continuous state space $\langle x, \dot{x}, \theta, \dot{\theta} \rangle$ : cart position, cart velocity, pole angle and pole velocity, and a discrete action space consisting of 2 actions: left and right. For a like-with-like comparison across standard learning approaches, we discretized the state-space into bins of $(3, 3, 6, 3)$.

We investigate agent intention in cartpole learnt from vanilla Q-learning and DQN. In both methods, we train the agent until it is reasonably stable. The vanilla Q-learning agent is trained with learning rate $\alpha = 0.1$ and $\gamma = 1$. The DQN agent is trained using Adam [17] with gradient clipping at $[-1, 1]$ and $\alpha = 0.0001, \gamma = 1$, while the belief map is generated by a deep network which we refer to as a *Deep Belief Network* (DBN) trained using the same hyperparameters.

Figure 4 shows the belief map versus the actual trajectory of the episode produced by forward simulation of the agent. The mismatch between forward simulation and expected states is particularly interesting. We believe this is because cartpole is chaotic but deterministic. Rerunning the same forward simulation results in exactly the same behaviour each time, however, as the agent sees a quantised approximation of the initial state, it has much greater uncertainty regarding future states.

**Taxi** [11] is a challenging environment where an agent must first navigate from a randomised start point, around obstacles, to collect a passenger from a random location, then pick up that passenger and navigate to a randomised destination. We investigate agent intention learnt by Q-learning and DQN. The Q-learning agent is trained with $\alpha = 0.4, \gamma = 1$ using a 4-tuple state representation of car x and y location, passenger location and destination. DQN used $\alpha = 0.0001, \gamma = 1$ for $100k$ episodes.

Figure 5 and Figure 6 shows visualisations of the RL agents' belief maps at several steps along a trajectory (animations are available in the supplementary material). For both methods we can intuitively reason about the intentions of the agents. These intentions become clearer over time, as the multiple potential routes (including the outward and return journey) which are overlaid, collapse to the single shortest path towards the goal.

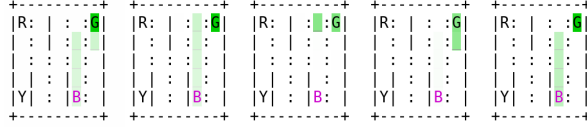

(a) $a^0$: move east, $a^1$: move south. $Q(s, a^0) = 7.7147$, $Q(s, a^1) = 7.0777$.

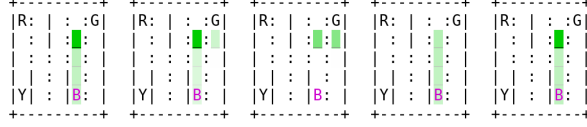

(b) $a^0$: move south, $a^1$: move east. $Q(s, a^0) = 11.87$, $Q(s, a^1) = 7.6599$.

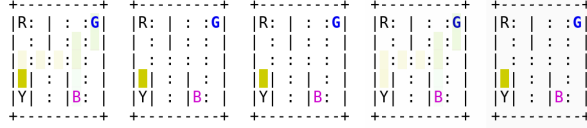

(c) $a^0$: move north, $a^1$: move east. $Q(s, a^0) = -2.3744$, $Q(s, a^1) = -5.204$.

Figure 7: Contrastive explanations of taxi tabular Q-learning agent during the episode in Figure 5. Here we denote $a^0$ as the best action, $a^1$ as the second best action. Current state is highlighted by the strong hue of the car. From left to right we have: (1) $\mathbf{H}(s, a^0)$, (2), $\mathbf{H}(s, a^1)$, (3) $\min(0, \mathbf{H}(s, a^1) - \mathbf{H}(s, a^0))$, (4) $\min(0, \mathbf{H}(s, a^0) - \mathbf{H}(s, a^1))$, (5) the intersection of the two expected outcomes, given by $\min(\mathbf{H}(s, a^0), \mathbf{H}(s, a^1))$.

**Contrastive Explanations**   Figure 7 demonstrates the capability of our method to extract contrastive explanations. We ask "What change in future states does the agent believe will happen when one action is selected over another?" To generate these explanations, we scale the belief map to $[0, 1]$. Figure 7a shows a real example of the value ambiguity problem of Theorem 1. By contrasting the belief maps associated with the best and second best actions we see two distinct but equally good policies were discovered. Such explanations is not possible with vanilla Q-learning.We also identify a failure case in Figure 7c where taking the suboptimal action would cause the car to be stuck in the same state. Even for DQN, although the mathematical guarantee is lost, we extract an approximation of the agent's intention; see supplementary material for contrastive explanations for the DQN agent.

## 6   Conclusions and Future Work

We have proposed a novel approach for explaining what outcomes are implicitly expected by reinforcement learning agents. We proposed a meaningful definition of intention for RL agents and proved no post-hoc method could generate such explanations. We proposed modifications of standard learning methods that generate such explanations for existing RL approaches, and proved consistency of our approaches with tabular methods. We further showed how it can be extended to deep RL techniques and demonstrated its effectiveness on multiple reinforcement learning problems. Finally we provided code to allow future researchers to introspect their own RL environments and agents.

A noticeable limitation of our method is that it is best suited for problems where tabular reinforcement learning works well (i.e. low-dimensional and easily visualisable). Just as deep learning substantially increased the applicability of reinforcement learning it is interesting to ask how it could increase the applicability of our approach. One potential answer lies in the use of concept activation vectors [16], which allows for shared concepts between humans and machine learning algorithms. For example, if a concept was trained to recognise a pinned piece in chess our approach would be able to explain a move by saying that it will give rise to a pinned knight later. Importantly, given a set of concepts $F$ defined over the state space $\mathcal{S}$, it would be possible to train the belief map over the low-dimensional space $F(\mathcal{S})$, rather than $\mathcal{S}$, substantially improving the scalability of our approach. As such, while our approach is directly applicable to problems with easily mappable state-spaces, it also lays the mathematical foundations for explainable agents in more complex systems.

## Acknowledgements

This work was partially supported by the UK Engineering and Physical Sciences Research Council (EPSRC), Omidyar Group and The Alan Turing Institute under grant agreements EP/S035761/1 and EP/N510129/1, and was directly supported by Vice-Chancellor's studentship award by the University of Surrey.

## Broader Impact

By making important early steps into explainability for Reinforcement Learning, this work has strong potential impact across a broad range of areas from autonomous vehicles to medical robotics. Beneficiaries of this work include researchers and developers who wish to gain additional insight into their agents behaviour, in order to improve performance. Additionally, if accidents arise in a deployed system, our work provides a mechanism to introspect these and attempt to prevent the same thing from happening in the future.

A failure in the proposed system could lead to an incorrect intention being attributed to an agents behaviour. Our proofs show that this cannot happen for table-based Q-learning, but it is potentially possible for DQN. An incorrectly attributed intention could potentially lead to unnecessary or harmful remedial action being taken to correct the agents behaviour. In extreme cases a failure of the system may lead to accountability being incorrectly attributed. Therefore it is vital to keep in mind the limitations of the theoretical guarantees provided with this work.

Our technique does not rely on any dataset biases. However the generated explanations are specific to a single instance of an agent's policy. There are no guarantees about generalization of intention between agents, even those which are similarly trained.

## Footnotes

*Part of this work was done while at the Alan Turing Institute and University of Surrey.

[1] https://github.com/hmhyau/rl-intention

[2]This is particularly common in multi-agent systems, where the behaviour of other agents can change.

[3]The reasoning in this paper holds true for estimation of state value function $V^{\pi}(s_t)$ as well, but for conciseness we focus on Q-functions.

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
