[Supplementary Material]

# Supplementary Materials for 'What Did You Think Would Happen? Explaining Agent Behaviour through Intended Outcomes'

**Herman Yau**
CVSSP, University of Surrey

**Chris Russell**
Amazon Web Services

**Simon Hadfield**
CVSSP, University of Surrey

Section 1 provide the definition of double Q-learning, the update equation for estimating its belief map, and a formal proof of consistency between the two. Section 2 provides additional experimental and architectural details about the implementations of our various techniques. Section 3 reports our findings for the Monte Carlo variants of our algorithm, in both the blackjack and cartpole environments. We also show additional exploration and insight into the contrastive explanations developed in the report.

## 1 Double Q-learning

Double Q-learning[1] is a modification of Q-learning which maintains two Q-tables making it is less prone to over-estimation. After each episode, one of the Q-tables ($Q^A$ or $Q^B$) is randomly updated, using one of the following two equations.

$$Q_i^A(s_t, a_t) \leftarrow (1-\alpha)Q^A(s_t, a_t) + \alpha \left( r_t + \gamma Q_{i-1}^B \left( s_{t+1}, \arg\max_a Q_{i-1}^A(s_{t+1}, a) \right) \right)$$

$$Q_i^B(s_t, a_t) \leftarrow (1-\alpha)Q^B(s_t, a_t) + \alpha \left( r_t + \gamma Q_{i-1}^A \left( s_{t+1}, \arg\max_a Q_{i-1}^B(s_{t+1}, a) \right) \right)$$

**Belief Map Update Step**  Similarly, we maintain two belief maps ($\mathbf{H}_A$ and $\mathbf{H}_B$) and update them in sync with the Q-tables, that is each time $Q_A$ is updated, we also update $\mathbf{H}_A$ and the same for $Q_B$ and $\mathbf{H}_B$. The updates are given by the following two equations

$$\mathbf{H}_i^A(s_t, a_t) \leftarrow (1-\alpha)\mathbf{H}^A(s_t, a_t) + \alpha \left( 1_{s_t, a_t} + \gamma \mathbf{H}^B \left( s_{t+1}, \arg\max_a Q^A(s_{t+1}, a) \right) \right)$$

$$\mathbf{H}_i^B(s_t, a_t) \leftarrow (1-\alpha)\mathbf{H}^B(s_t, a_t) + \alpha \left( 1_{s_t, a_t} + \gamma \mathbf{H}^A \left( s_{t+1}, \arg\max_a Q^B(s_{t+1}, a) \right) \right)$$

**Proof of Consistency**

*Proof.* The proof is almost identical to that of Q-learning in the main body of the paper. In the base case, all $Q$ and $\mathbf{H}$ are zero initialised, and therefore consistent. For the inductive step, after each episode, one of $A$ or $B$ is updated, and we assume that both $Q_{i-1}^A$ and $Q_{i-1}^B$ are consistent with $\mathbf{H}_{i-1}^A$ and $\mathbf{H}_{i-1}^B$. Without loss of generality, we assume $A$ is selected: We consider time $i$ and we write $m_t = \arg\max_{a \in \mathcal{A}} Q_{i-1}^A(s_t, a)$ then:

$$
\begin{aligned}
\text{vec}(\mathbf{H}_i^A(s_t, a_t))^\top \text{vec}(\mathbf{R}) &= \text{vec}\left( (1-\alpha)\mathbf{H}_{i-1}^A(s_t, a_t) + \alpha \left( 1_{s_t, a_t} + \gamma \mathbf{H}_{i-1}^B(s_{t+1}, m_{t+1}) \right) \right)^\top \text{vec}(\mathbf{R}) \\
&= (1-\alpha)Q_{i-1}^A(s_t, a_t) + \alpha \text{vec}\left( 1_{s,a} + \gamma \mathbf{H}_{i-1}^B(s_{t+1}, m_{t+1}) \right)^\top \text{vec}(\mathbf{R}) \\
&= (1-\alpha)Q_{i-1}^A(s_t, a_t) + \alpha(r_t + \gamma Q_{i-1}^B(s_{t+1}, m_{t+1})) \\
&= Q_i^A(s_t, a_t).
\end{aligned}
$$

as required.  □

## 2 Agent Description

**Blackjack** In both Monte Carlo control and Q-learning we share the same training settings. We set the learning rate $\alpha = 0.1$, discount factor $\gamma = 1$. We set an initial exploration probability $\epsilon = 1$ which is exponentially decreased to $\epsilon = 0.05$ with a decay rate of $0.9999$ throughout the training.

**Cartpole** Similar to blackjack, we share the same training settings in cartpole. We set learning rate $\alpha = 0.1$ and discount factor $\gamma = 1$. Episode terminates when the length of the episode reaches 200 timesteps. We initially set exploration probability $\epsilon = 1$ which is linearly decreased to $\epsilon = 0.1$ throughout the first 500 episodes.

In DQN training, we use Adam [2] optimizer with $\epsilon = 1e - 8$, exponential decays $\beta_1 = 0.9, \beta_2 = 0.999$. The learning rate is $\alpha = 0.0001$. We use Huber loss with discount factor $\gamma = 1$. We clip gradients to be in the range of $[-1, 1]$. For each learning iteration, we batch 16 experience together for optimisation. We initially set exploration probability $\epsilon = 1$ which is linearly decreased to $\epsilon = 0.1$ throughout the first 500 episodes. Full details of the neural network architectures can be found in Table 1 and 2.

| Layer | Type | Input | Size |
|---|---|---|---|
| input | N/A | N/A | 4 |
| fc1 | MLP | input | 128 |
| fc2 | MLP | fc1 | 512 |
| output | MLP | fc2 | 2 |

Table 1: DQN neural network architecture

| Layer | Type | Input | Size |
|---|---|---|---|
| input | N/A | N/A | 162 |
| fc1 | MLP | input | 512 |
| fc2 | MLP | fc1 | 1024 |
| fc3 | MLP | fc2 | 2048 |
| output | MLP | fc3 | 2*2*162 |

Table 2: DBN neural network architecture. Output is reshaped to $\mathbb{R}^{A \times S \times A}$ before return.

**Taxi** We first describe the training details for Q-learning. We set the learning rate $\alpha = 0.4$ and discount factor $\gamma = 0.9$. An episode terminates when the agent completes the episode or reaches a threshold of 200 timesteps. We initially set exploration probability $\epsilon = 1$ which is linearly decreased to $\epsilon = 0.1$ in the first 250 episodes.

In DQN training, we use Adam [2] optimizer with $\epsilon = 1e - 8$, exponential decays $\beta_1 = 0.9, \beta_2 = 0.999$. The learning rate is $\alpha = 0.0001$. We use Huber loss with discount factor $\gamma = 1$. We clip gradients to be in the range of $[-1, 1]$. For each learning iteration, we batch 16 experience together for optimisation. We initially set exploration probability $\epsilon = 1$ which is linearly decreased to $\epsilon = 0.1$ in the first 250 episodes. In order to speed up training of DBN, we made a small modification by splitting belief update and action into two inputs: belief update is a binary indicator function 1 at position $s$, and action is converted into one-hot encoding before being fed into the neural network. Full details of the neural network architectures can be found in Table 3 and 4.

| Layer | Type | Input | Size |
|---|---|---|---|
| input | N/A | N/A | 4 |
| fc1 | MLP | input | 500 |
| fc2 | MLP | fc1 | 2000 |
| output | MLP | fc2 | 6 |

Table 3: DQN neural network architecture

| Layer | Type | Input | Size |
|---|---|---|---|
| input_belief | N/A | N/A | 500 |
| input_action | N/A | N/A | 6 |
| belief_stream | MLP | input_belief | 1024 |
| action_stream | MLP | input_action | 128 |
| concat | N/A | input_belief, input_action | 1024+128 = 1152 |
| fc3 | MLP | concat | 2048 |
| output | MLP | fc3 | 500 |

Table 4: DBN neural network architecture. Output is reshaped to $\mathbb{R}^{A \times S \times A}$ before return.

## 3 Further Results

Suppose we have the best action $a^0$ and second best action $a^1$ at state $s$, we can compute a contrastive explanation $G$ by subtracting the belief of $a^0$ against $a^1$:

$$G(s, a^0, a^1) = \mathbf{H}(s, a^0) - \mathbf{H}(s, a^1) \tag{1}$$

Since $\mathbf{H}(s, a)$ is a decomposition of $Q(s, a)$, the subtraction will tell us which future states constitute the overall goodness of $a^0$ over $a^1$ in $s$. We apply equation 1 to generate the contrastive explanations below.

**Blackjack**   For conciseness, the visualisations of blackjack have been trimmed to only show reachable states. Figure 1 shows a belief map computed for an agent trained using Monte-Carlo control. This can be compared against that of the tabular Q-learning agent in Figure 3 of the main paper.

In Figure 2 and Figure 3 we verify that we can recover the Q-table from learned beliefs via a proxy reward map, thus our theorem in the main paper holds. In figure 4 and 5 we show that different contrastive explanations can be extracted by applying equation 1.

**Cartpole**   We verify in figure 7 and 8 that our theorem in the main paper holds without the aid of a proxy map. We also give a demonstration of contrastive explanations in figure 9 and 10.

**Taxi**   We provide animations of Figure 4 and 5 in the main text; see attached video for the animation. We also give contrastive explanations from DQN agent in Figure 11. Although the contrastive explanations are noticeably fuzzier, the extracted intention is similar to our results for Q-learning.

(a) Belief map for sticking   (b) Belief map for hitting   (c) Reward belief for sticking

(d) Reward belief for hitting

Figure 1: Monte Carlo control blackjack simulation: here we show the visualisations of belief maps and proxy action-reward map when player sum = 10, dealer card = 7 with no usable ace.

Figure 2: Q-learning: visualisations of ground truth Q-table and recovered Q-table from learned belief.

Figure 3: Monte Carlo control: visualisations of ground truth Q-table and recovered Q-table from learned belief.

(a)

(b)

Figure 4: Monte Carlo control, blackjack simulation: contrastive explanation when player sum $= 10$, dealer card $= 7$ with no usable ace. Figure 4a shows that hitting would grant access to various future states. This is reflected in the figure 4b, as the red block (consequence of sticking) indicates sticking is undesirable since the action is more likely to generate $-1$ reward.

(a)

(b)

Figure 5: Monte Carlo control, blackjack simulation: contrastive explanation when player sum $= 17$, dealer card $= 9$ with no usable ace. In contrast to figure 4a, hitting is more likely to yield a more negative consequence than twisting, as evidenced by 5a that the red block (consequence of hitting) at $(1, -1)$ clearly outweighs the cumulative sum of blue blocks (consequence of sticking).

(a) Belief map

(b) Forward Simulation

Figure 6: Monte Carlo control, cartpole simulation: belief map and trajectory visualisations.

Figure 7: Monte Carlo control, cartpole simulation: Q-table and recovered Q-table from learned beliefs. Each value is numerically identical except for floating-point errors.

Figure 8: Q-learning, cartpole simulation: Q-table and recovered Q-table from learned beliefs. Each value is numerically identical except for floating-point errors.

Figure 9: Monte Carlo control, cartpole simulation: selected contrastive explanations at state 79. We can intuitively reason and justify the agent's decision of moving to the right since it offers a greater stability. Moving to the left risks heading towards the edges highlighted by the red blocks.

Figure 10: Q-learning, cartpole simulation: selected contrastive explanations at state 79. The figure shows that moving to the right is undesirable since it will lead the agent to visit states which can cause possible failure, whereas moving to the left can offer more guarantee of staying in the middle.

(a) Value ambiguity. $a^0$: move south, $a^1$: move west. $Q(s, a^0) = 14.9908$, $Q(s, a^1) = 14.9858$.

(b) Bounce back. $a^0$: move south, $a^1$: move west. $Q(s, a^0) = 17.9323$, $Q(s, a^1) = 15.9981$.

Figure 11: DQN, taxi simulation: contrastive explanations of taxi DQN agent. Refer to Figure 7 in main text for descriptions of each subplot.