[Reviews · NeurIPS 2020]

Review 1

Summary and Contributions: The paper proposes an explanation method for reinforcement learning, in which the explanation consists of the series of actions the agent intends to take from a give state-action pair (they call these belief maps). Authors show that this type of explanation cannot be obtained from Q-functions alone, and it requires keeping track of additional information during Q-learning. Their proposal consists of keeping track of the expected states-rewards visitations matrices as well, which can substantially add to the memory and computational resources needed for Q-learning.

Strengths: Given the increasing use of RL in real-world applications that affect people, providing usable explanations for RL is indeed an important and timely research problem. I appreciate the authors' focus on the concept of intention, and the (brief) connection they make to the social pscychology literature. As I explain below, the theoretical claims are sound and empirical illustrations are helpful in conveying the method.

Weaknesses: The authors briefly mention several real-world applications that can motivate their work (e.g., autonomous driving, medical robots). I wish they had mentioned these motivating cases earlier on (in the introduction) and in greater detail. Also, I believe it is important for the authors to clarify early on whether they focus on local or global explanations. Most importantly, who is the target audience of the proposed explanations? Is it RL practitioners or non-technical users of the RL model? Either way, I believe the authors should provide evidence regarding the usefulness of their proposed method for the target audience. As authors acknowledge, the proposed method is only applicable to settings where |S|x|A| is not huge. While authors outline in their concluding remarks how their approach can be utilized in settings with large state-action spaces, an empirical illustration of whether the proposal works in such settings, and what can potentially go wrong would strengthen the work.

Correctness: Authors provide easy-to-follow proofs for their main results---which appear correct to me. They provide several empirical illustrations of their approach on standard RL datasets (OpenAI Gym, Blackjack, Cartpole, and Taxi), but they don't compare their approach with prior proposals, such as [Juozapaitis et al. 2019] in terms of their practical usability for the target audience.

Clarity: The paper is well-written and easy to follow. There are, however, several minor modifications the authors can make to improve accessibility to a broader audience. * Specify what acronyms such as DQN, PIRL stand for. * In equation (1), the notation \pi^*(a|s) is confusing. \pi^* is supposed to be a function from S to A. * In equation (5), \theta^- has not been defined. * In section 4, contrastive explanations paragraph: authors seem to have a specific interpretation of contrastive explanations in RL (i.e., contrasting the intentions of two actions). The term "contrastive explanation" often refers to a broad range of explanations methods. If authors mean to say that their method can be thought of as an *instance* of this class, I'd suggest they clarify this.

Relation to Prior Work: Authors clearly explain the difference between their approach and previous RL explanation methods in their related work section, but they don't provide an empirical comparison.

Reproducibility: No

Additional Feedback: The code to reproduce the results was missing, although authors claim they provide the code on their webpage. ----Update after reading authors' feedback------- Authors have responded to most of my concerns to a satisfactory degree and therefore, I raise my score. In particular, I think the example provided in the response can simultaneously clarify several issues: how does the explanation method compare with prior ones, how can the explanations be useful in practice, and what would the explanation look like concretely. I urge the authors to include this example in the main body of the paper.


Review 2

Summary and Contributions: Update: Thanks for your response. I actually really liked the idea in this paper, and I appreciate your inclusion of figure 1 in your response, but I have 2 main issues that I think would take a substantial re-writing and more experiments to address. 1) I really didn't get from the writing that the mismatch between the agent's (implicit) model and the true environment is a major part of the paper. (Without this piece, I don't know what your method provides over forward simulating the agent's behavior, which is a pretty easy thing to do.) 2) I think really demonstrating that this is an important issue requires some more experiments. Figure 1 shows that cases can be engineered where the agent's internal model does differ from what would happen in the actual environment and that your explanation can highlight that, but I don't think it's enough to show that it's an issue in realistic scenarios. I agree with you that it probably is, but it's hard to know without more experiments. --- This paper presents an approach to explaining reinforcement learning agents through the agent's intended consequences. Through this, they aim to answer the question of what the agent thought would happen in the future when deciding on a particular action. They provide a proof demonstrating that an agent's belief map, i.e. the discounted states it expects to visit in the future, cannot be uniquely determined from the Q function. They then provide a procedure for learning the agent's belief map alongside the Q function in a way that is consistent with the Q function.

Strengths: Explaining reinforcement learning agents is an important problem, and I think explaining them in terms of what future events the agent believes will happen is a good idea! I also really liked the idea that if the post-hoc explanation is underspecified, perhaps the explanation needs to be learned alongside the agent to fully capture the nuances of its behavior.

Weaknesses: One of the key ideas wasn't clear to me: how does this kind of explanation compare to forward simulating under the agent's learned policy and recording which states are visited? Is the hypothesis that the agent's "mental model" of the true transition function is flawed and that knowing this is useful information? (In the batch setting, you may not have the transition function, but you could build a model of it based on your data. How much would this approximation affect the resulting explanation?) I could imagine there being important subtleties between these 2 approaches, but I would have found it helpful to have those hypotheses laid out clearly in the introduction, and tested in the experiments. There were a few things I thought were missing from the related work. The first is this paper: https://arxiv.org/pdf/1807.08706.pdf that tackles a somewhat similar problem. How does your approach compare to this? I was also curious how these belief maps compare to the successor representation: http://www.gatsby.ucl.ac.uk/~dayan/papers/d93b.pdf and the work building on it. Finally, I have several comments and questions about the experiments. As I mentioned above, I would like to see how this approach differs from inspecting simulations of the agent's behavior from a particular starting state + action. Generally, I would have also found it helpful to have more explicit conclusions drawn from the results. I also have some more minor questions. - In the blackjack setting, why doesn't the dealer's hand change? Is it just fixed and if so, why? I also wasn't sure how to interpret Figures 2c and 2d. - In the cart pole domain, what does it entail that the DQN estimates are fuzzier? Can we interpret anything from these beyond that the agent is doing something reasonable? It would be helpful to have examples where the belief maps show something surprising about the agent's behavior or something it would have been hard to identify without the belief map. - Finally, in the Taxi domain, I don't see the bias that the DQN agent exhibits in the figure. What should I be looking for? I think figure 6 may have a typo--descriptions for columns 3 and 4 look to be the same. I would also find it helpful to have an English description of what these columns mean. I also didn't understand why the figure in row b, 4th column has the trajectory that both agents visit highlighted. I would have expected it to have nothing highlighted since there are no squares visited in the column 1 figure but not the column 2 figure.

Correctness: For the proof of theorem 1, I didn't see how this would imply that it is never possible to produce a post-hoc interpretation given multiple optimal policies. I read this as just giving one counterexample to demonstrate that this is not always possible. Did I miss something here? Even if my interpretation is correct, I think this would still cast doubt on the usefulness of a post-hoc explanation of this kind.

Clarity: I found the main idea of the paper confusing and I think some re-writing to clarify the points I mentioned in the weaknesses would make it easier to understand. Other than that, I found the structure of most of the paper relatively easy to follow. In the results, I would have liked to see more details about how to read and interpret the results, as well as more explicit conclusions.

Relation to Prior Work: There are a few things I would have found it helpful to see compared/contrasted. I mentioned these in weaknesses.

Reproducibility: Yes

Additional Feedback:


Review 3

Summary and Contributions: This paper presents an approach to help explain the `intention' of an RL agent by collecting additional information during training. Equipped with this new information, one can form a trajectory of belief states that capture and perhaps provide agent intention. The paper provides a theorem to show that the explanation is consistent with the Q(s,a) estimate. Overall, the approach might allow further introspection and debugging of agent behavior.

Strengths: This paper is well motivated and tackles an under explored domain in interpretability. Most recent work in the interpretability area has focused on classifiers or perhaps generative models. Here the authors provide a new insight and approach for interpretability of RL agents. The first key strength of this paper is an interesting and different attack on an under explored problem. The paper also provides comprehensive assessment across different environments like pong, blackjack, and taxi. Here the belief trajectory seems to show interesting behavior for the agents. The 'intentionality' principle that motivates this work is quite interesting and hasn't been previously explored in the literature to my knowledge. Theorem 1, though straightforward, might have quite profound implications for how RL agents ought to be interpreted. The 'explanation' function 'H' defined in this work is in some sense analogous to a Q function, so H is updated in a manner identical to how an H function is also updated.

Weaknesses: While overall an interesting submission, I am still quite tentative as to what to take away from the empirical interpretations. The nature of my qualms regards how to falsify/confirm several of insights presented in that section. For example, can these insights be used to alter the behavior of an agent? If yes, then such intervention might show that the insights discussed in these sections reveal true behavior. I go into more detail on this issue in the additional feedback section.

Correctness: The theorems presented are correct (as far as I can tell). The approach presented is also correct since the authors show consistency for the H function. However, it is unclear how to verify the insights discussed in the empirical section.

Clarity: The paper is clear and well written. It is free of typos, and grammar problems.

Relation to Prior Work: The sum total of work in this area is relatively small. However, this paper does a good job of providing context on recent work on explainability in RL and how the approach presented here differs from these previous works. In general, I think the paper shows familiarity with the necessary work in this area.

Reproducibility: Yes

Additional Feedback: In this section I will go into detail about the perceived weakness that I mentioned above. Validating Empirical Insights. One issue that plagues work on interpretability is the problem of validating agent/model insights. The belief map is interesting, but I am hoping the authors can provide further evidence that these belief maps indeed demonstrate agent `intent'. Is it possible to design an agent whose intent is known a priori so that you can then compare the observed behavior to the maps that your method produces? I am not sure how this would work since the additional information for your method is also collected during training. Another way would be to use the insight from the maps to somehow change the agent behavior after observing the belief maps. This could show evidence that the maps reflect agent behavior. The upshot here is that I am hoping to understand or get at a way to sanity check the insights that we learn from the maps presented. More discussion on motivation It would be useful for the authors to discuss why the formulation presented is desired beyond other kinds. In addition, it would be useful to answer the question, what kind of interpretation can this method not provide? I am trying to get at the limits of the work here. Perhaps a specific question about what I am hoping for, why is p(s_t+n | s_t, a_t, \pi) useful for understanding the agent intent as supposed to say, p(a_t+n | s_t, a_t, \pi) (assuming this is even a computable quantity)? Minor Comment In paragraph 2, the authors say that there two groups of interpretation methods; this is not the case. Even if we restrict attention to the case of deep networks it is still not the case. There are 1) attribution methods (as was methods, i.e. saliency maps etc), 2) exemplar/input ranking methods like the work on influence functions etc, 3) there are concept methods like TCAV, and 4) there are methods that design the model class to be interpretable by design. This list is not exhaustive, but the discussion there should be amended or qualified based on the interpretation that the authors are going for. line 114 to predict to future -> to predict the future Post-rebuttal ------------------------------ Thanks to the authors for the responses and clarifying my questions. I implore the authors to respond to the issues raised by R2 on more clearly comparing their setup to simulating the agent under the learned policy.


Review 4

Summary and Contributions: The authors provide approaches for explainable RL, where information needed for explanations is collected during training. They demonstrate their approach on different RL problems. They propose a deocomposition of the Q-function over state and action space, giving detailed reasons of sub-rewards over future states. This is an addition to standard RL frameworks.

Strengths: This paper attempts to address the issue of black-box models, but enabling explainable RL. While current explanations show what in the environment drives agents to take action, the authors aim to show what the agents expects to achieve as a result of an action choice (intent-based explanations).

Weaknesses: None, however I am not an expert in RL

Correctness: The methods seem sensible and the experiments chosen make sense and are descriptive of the approach.

Clarity: The paper is well-written and clear.

Relation to Prior Work: The authors have satisfactorily covered current methods of interpretability for RL models, and pointed out where their work fits in.

Reproducibility: Yes

Additional Feedback:

[Author Response · NeurIPS 2020]

We thank the reviewers for their constructive feedback and suggestions to strengthen this work. All points raised will be addressed in the revised version of the manuscript.

**R1:** As suggested we will include motivating examples in the introduction (see new fig under **R2**). We will emphasise our explanations are local (what is expected to happen given the current state and action), and that our target audience is RL practitioners trying to understand agents. We will ensure that acronyms and mathematical notations are specified. This includes changing $\pi^*(a|s)$ to $\pi^*(s)$ and explaining that $\theta^-$ is the target network parameters. We will also clarify the instance of our chosen contrastive explanation.

*HCI Comparison with [10].* Our approach shows what the agent expects to be the consequences of the action while [10] highlights the components of the environment that lead an agent to select an action. These are complementary explanations, and we feel that a user study comparison on any particular task would simply show if information about the environment or future state is more relevant to that task.

*Large state-action spaces and potential pitfalls.* This is a suggestion for potential future research which was mentioned in the conclusion. Unfortunately, combining concepts with RL explanations will require significant further work.

*Code is missing.* Apologies for any confusion. We meant to say we will provide the full code on publication. We could not make our code repository public without breaking the anonymity of the review process.

**R2:** *In blackjack why doesn't the dealer's hand change?* The dealer shows one card, and then the agent plays in response. The agent learns that once played, the revealed card never changes, as shown by the belief maps.

*Interpreting figure 2c and 2d.* These are four terminal states created to make the reward function deterministic: *lose*, *win*, *bust*, and *draw* and one transition state *hit, no bust*. The figure will be updated so the states are clearly labelled.

*Compare your explanations to forward simulation.* This is a great idea. N.B. the unpublished work [van der Waa et al., 2018] performs these forwards simulations. We have added Fig 1 to the paper to illustrate these differences. It shows the belief of a blackjack agent mistrained on a small and biased replay buffer. This leads the agent to believe that hitting on a 15 always leads to a 20. Meanwhile the forward simulation on the right shows the "real" behaviour of the system and therefore provides no insight into problems at training time.

We thank the reviewer for directing us to Dayan [1993], which is closely related, albeit outside the context of XAI. We will discuss this carefully in the final version.

Figure 1: Belief map (left) vs forward simulation (right).

*How theorem 1 implies it is never possible to produce a post-hoc interpretation.* Theorem 1 shows that a general explainer can not always generate correct post-hoc interpretations, by showing an ambiguous example where it fails. The reviewer is correct, it is sometimes possible to give correct post-hoc interpretations and there is always a trivial explainer that exactly describes the behaviour of any given agent and no others. In contrast theorem 2 guarantees our explainer is always correct.

*Fuzzy DQN belief estimates in cartpole* We suspect the fuzzy estimates may be due to: (1) low learning rate; the sparse belief map updates could adversely affect the magnitude of gradients (also present in tabular belief map trained at a very low learning rate). (2) distribution of the experience replay buffer; poor experiences can cause a shift in the agent which is exacerbated by batch learning. This is another example of the insights provided by our method.

*Bias not present in Figures 4 and 5.* This text was accidentally left in from an earlier draft with different results. You are correct that the current figures do not show any such bias.

*Why isn't col 4 of fig 6(b) blank since col 1 and 2 match.* Although every cell visited in subfig 1 is also visited in subfig 2, the strengths of the belief in those states differs due to the discount factor of future states ($\gamma = 0.9$). Also note the caption for column 3 should read $\min(0, \mathbf{H}(s, a_1) - \mathbf{H}(s, a_0))$.

**R3:** *Evidence that the belief maps indeed demonstrate the agent's 'intent'.* The new Fig. 1 (see R2.3) shows the intention of a mistrained agent. Moreover, theorem 2 is a proof that the explanations really are consistent with what the agent expects to happen, and the result has also been validated numerically.

*Why the formulation is desired beyond other kinds?* We don't claim that our formulation is always more desired, however it provides a new type of information no other method offers. These explanations are more like those psychologists say people offer (see intro) and can be used to identify a gap between replay buffers and the true environment (see fig. 1).

*Why is $p(s_t + n|s_t, a_t, \pi)$ useful to understand the agent intent compared to $p(a_t + n|s_t, a_t, \pi)$?* It is simple for our method to compute the latter. However, we feel it provides less insight: In taxi it would show how often the taxi moves in each direction over an entire episode, while in blackjack it would show the average number of sticks and hits. Meanwhile our approach returns the route of the taxi, and the sequence of card values.

*It is incorrect to say there are just two groups of interpretation methods.* Yes this is an oversimplification. We will clarify.

**R4:** The reviewer did not ask for any clarifications in the rebuttal. Nevertheless we greatly appreciate the kind words and the affirmation of the importance of this research.

[Meta-Review · NeurIPS 2020]

This paper tackles an interesting problem of explaining agent behavior in RL and doing so in the form of future events. The theoretical claims, experiments, and writing of this paper are done well. However, the paper suffers from the following drawbacks: 1. Writing about certain assumptions is somewhat unclear -- e.g., the assumption that there exists a mismatch between agent's model and true environment -- this is a major underlying assumption which is not clarified upfront. It would also be good to show that this assumption indeed holds in the real world with some experiments. 2. There is no clarification on whether the resulting explanation is local or global until much later in the paper.